



# Unfolding the relationship between seasonal forecast skill and value in hydropower production: A global analysis

Donghoon Lee[1,2], Jia Yi Ng[3], Stefano Galelli[3], and Paul Block[1]

[1]Department of Civil and Environmental Engineering, University of Wisconsin-Madison, Madison, Wisconsin, USA.
[2]Climate Hazards Center, Department of Geography, University of California, Santa Barbara, California, USA.
[3]Pillar of Engineering Systems and Design, Singapore University of Technology and Design, Singapore.

**Correspondence:** Stefano Galelli (stefano_galelli@sutd.edu.sg)

**Abstract.** The potential benefits of seasonal streamflow forecasts for the hydropower sector have been evaluated for several basins across the world, but with contrasting conclusions on the expected benefits. This raises the prospect of a complex relationship between reservoir characteristics, forecast skill and value. Here, we unfold the nature of this relationship by studying time series of simulated power production for 735 headwater dams worldwide. The time series are generated by

running a detailed dam model over the period 1958–2000 with three operating schemes: basic control rules, perfect forecast-informed, and realistic forecast-informed. The realistic forecasts are issued by tailored statistical prediction models—based on lagged global and local hydro-climatic variables—predicting seasonal monthly dam inflows. As expected, results show that most dams (94%) could benefit from perfect forecasts. Yet, the benefits for each dam vary greatly and are primarily controlled by the time-to-fill and the ratio between reservoir depth and hydraulic head. When realistic forecasts are adopted, 25% of

dams demonstrate improvements with respect to basic control rules. In this case, the likelihood of observing improvements is controlled not only by design specifications but also by forecast skill. We conclude our analysis by identifying two groups of dams of particular interest: dams that fall in regions expressing strong forecast accuracy and have the potential to reap benefits from forecast-informed operations, and dams with strong potential to benefit from forecast-informed operations but fall in regions lacking forecast accuracy. Overall, these results represent a first qualitative step towards informing site-specific

hydropower studies.

## 1 Introduction

Hydropower is the leading form of renewable power, contributing to 16% of global electricity production and 62% of all renewable electricity generation (IHA, 2019). Total hydropower production is expected to double by 2050, with substantial growth in Asia, Africa, and South America (Zarfl et al., 2015; Zhang et al., 2018). The sustainable operation of hydropower facili-

ties, however, is challenged by hydro-climatic variability, namely seasonal and inter-annual fluctuations in streamflow—and hydropower output—driven by large-scale climate drivers. Examples include the North Atlantic Oscillation (NAO), affecting hydropower in Europe (De Felice et al., 2018), or the El Niño Southern Oscillation (ENSO), affecting one third of the world's hydropower dams (Ng et al., 2017). An attractive management option to limit these fluctuations is the use of adaptive reservoir operating policies based on seasonal streamflow forecasts (Troin et al., 2021). Hydropower operators in snowmelt-dominated





regions, for instance, can rely on seasonal forecasts to commit to reservoir drawdown in early winter in preparation for future
inflows. Yet, the benefits reaped from such decision (forecast *value*) may vary in response to the forecast accuracy, or *skill*, as
well as the design specifications of the reservoir system at hand.

Perhaps unexpectedly, several studies have shown that using streamflow forecasts can lead to tangible gains (Kim and
Palmer, 1997; Block, 2011; Libisch-Lehner et al., 2019; Ahmad and Hossain, 2019), but also that these gains vary widely.
Maurer and Lettenmaier (2004), for instance, observed a modest 1.8% increase in hydropower production when re-operating
the reservoirs along the Missouri River with perfect forecasts. Similarly, Rheinheimer et al. (2016) found a 1.2% increase
in the economic gain for the Sierra Nevada's hydropower system. In contrast, Hamlet et al. (2002) estimated that seasonal
streamflow forecasts could raise hydropower revenue by \$153 million/year ($> 40\%$) in the Columbia River basin. How do we
explain such differences? To answer this question, we need to understand the relationship between forecast skill, value, and
reservoir characteristics. There are two common approaches for tackling this problem. In the analytical approach, one typically
uses synthetic forecasts and a hypothetical reservoir system to analytically derive a relationship between the aforementioned
variables. For example, You and Cai (2008) derived a theoretical relationship linking the ideal forecast horizon to various
factors, such as water stress level, reservoir size, or inflow uncertainty. In a follow-up study, Zhao et al. (2012) investigated
the relationship between forecast horizon and uncertainty, identifying an effective forecast horizon that balances the effects of
horizon and uncertainty, providing the largest benefit to the reservoir operators. On the other hand, the experimental approach
simulates the operations of existing reservoirs systems with seasonal streamflow forecasts to determine their potential value,
and, where possible, to build an empirical relationship linking forecast value, skill, and reservoir characteristics. Maurer and
Lettenmaier (2004), for instance, attributed the relatively low gains found for the Missouri River basin to the system's large
storage capacity (relative to annual inflow). When studying the Sierra Nevada's hydropower system, Rheinheimer et al. (2016)
found that forecast value is insensitive to storage capacity, yet highly sensitive to powerhouse capacity.

One limitation of the existing literature is that it illustrates the potential benefits of seasonal forecasts on individual hy-
dropower dams or specific river basins. In turn, this leads to a fragmented knowledge of how forecast skill and reservoir
characteristics translate into forecast value. A global-scale assessment of forecast skill and value can fill this gap and provide
numerous benefits. First, evaluating forecast skill and value for hundreds of hydropower sites can frame the current body of
knowledge within a larger scope, and elicit the wide range of possible benefits observed at individual dams. Second, a global
study offers a broad spectrum of actual reservoir characteristics, allowing for an in-depth analysis of how forecast value is mod-
ulated by reservoir characteristics. To date, this important aspect has been primarily demonstrated for water supply reservoirs
(Anghileri et al., 2016; Turner et al., 2017a), but remains largely unexplored for hydropower dams (Yang et al., 2021). Such
added knowledge may not necessarily lead to specific design guidelines, but may still be valuable to operators and planners.
For example, a quantitative relationship between reservoir characteristics, forecast skill, and value could be applied in regional
analyses designed to determine the minimum forecast skill required by an existing or planned reservoir network (Bertoni et al.,
2021). The expected continuous development of forecast systems will only continue to foster such analyses (Johnson et al.,
2019; Crochemore et al., 2020; Troin et al., 2021).





Here, we present a global analysis carried out on 753 headwater dams, representing 10% of the world's installed hydropower capacity. Specifically, we leverage recent studies demonstrating global streamflow predictability conditioned on large-scale climate variability (Ward et al., 2014; Lee et al., 2018), and develop seasonal inflow forecasts for each dam. Then, we quantify the value of these forecasts by comparing the amount of hydropower simulated by three operating schemes based on realistic forecasts (issued by our model), perfect forecasts, and (no forecasts) control rules. We leverage the wide range of climatic conditions and dam characteristics available in our database to (1) explain how reservoir design properties and forecast skill affect the value of seasonal forecasts, and (2) identify key geographical regions where dams can benefit most from forecasts. The relationships between forecast skill, value, reservoir characteristics, and geographic location revealed through these analyses represent a first step toward informing site-specific studies.

## 2 Data

### 2.1 Hydropower dams data

We use the database introduced by Ng et al. (2017) containing design specifications for 1,593 hydropower reservoirs— representing almost 40% of the world's installed hydropower capacity. The database provides information on dam height, storage capacity, maximum surface area, long-term average discharge, upstream catchment area, geographic coordinates, installed power capacity, maximum turbine flow, and operating goals (e.g., hydropower supply, flood control). The majority of these data are retrieved from the Global and Dam (GRanD) database (Lehner et al., 2011) and complemented with data from the International Commission on Large Dams (ICOLD, 2011), the Global Lakes and Wetlands Database (Lehner and Döll, 2004), and the Global Energy Observatory (GEO, 2016). We filter out all dams affected by upstream regulation, reducing the number from 1,593 to 753, representing head water-only dams. Filtering is based on the Degree Of Regulation (DOR) for each dam, defined as the ratio between the storage volume of the upstream dam(s) and the natural average discharge volume of a given river segment (Grill et al., 2019). We retain only dams with DOR values equal to 0.

To model the relationship between storage and depth, we use an approach commonly adopted in global studies (Van Beek et al., 2011; Turner et al., 2017b). Specifically, we model the storage-depth relationship with Kaveh's method, which assumes an archetypal reservoir shape (Kaveh et al., 2013). This method estimates the reservoir surface area as a function of volume, maximum surface area, depth, and maximum depth. For the limited number of cases in which maximum depth is not available, we adopt Liebe's method and assume that the reservoir is shaped like an inverted pyramid cut diagonally in half (Liebe et al., 2005). To test the accuracy of these assumptions, we infer the bathymetry of each dam from a high-resolution global hydrography dataset (Yamazaki et al., 2019), and retain for comparison approximately 200 reservoirs with dam height and storage capacity estimates comparable to those reported in the GRanD database (see Text S1). Results, reported in Figure S1, indicate that forecast value is not strongly affected by the approach adopted to model the storage-depth relationship.

For each dam, we obtain a monthly inflow time series from the Water and Global Change (WATCH) 20th century model gridded global runoff dataset (Weedon et al., 2011). The runoff data are generated by the global hydrological model WaterGAP (Alcamo et al., 2003), which estimates the accumulated runoff for each grid ($0.5° \times 0.5°$ resolution) using the DDM30 river





network (Döll and Lehner, 2002). The model is calibrated with discharge data from the Global Runoff Data Center and has been applied to many global water resources studies (Döll et al., 2009; Haddeland et al., 2014). However, the course spatial resolution may be a source of uncertainty for dams located in small catchments. For this reason, we modify the original WATCH database in three ways. First, we consider only the period 1958–2000, which contains more detailed forcing data (Weedon et al., 2011). Second, we manually adjust the position of 270 dams (of the 753 dams) to properly align with the DDM30 river network, using the HydroSHEDS river network (Lehner et al., 2008) and satellite images. Finally, we correct the discharge data to account for any disparity between the upstream catchment area defined by the DDM30 river network and the documented upstream catchment area of each dam (Ng et al., 2017).

Climate-specific information for each dam location is also obtained, based on the updated Köppen-Geiger climate classification, which is conditioned on global, long-term monthly precipitation and temperature time series (Peel et al., 2007). Specifically, we use the Köppen-Geiger climate classification that occurs most frequently in the grids upstream of each dam.

## 2.2 Hydro-climatological data

The seasonal forecasts developed here depend on seven potential predictors: four large-scale climate drivers (ENSO, NAO, Pacific Decadal Oscillation (PDO), and the Atlantic Multidecadal Oscillation (AMO)), and three variables accounting for local processes (lagged inflow, snowfall, and soil moisture). The four large-scale climate drivers are interannual, decadal, or multidecadal quasi-periodic oscillations derived from oceanic and atmospheric fields, and play a key role in determining hydro-climate patterns across the world (Lee et al., 2018). To characterize ENSO, we use the Niño 3.4 index, defined as the anomalies of 3-month running mean Sea Surface Temperatures (SST) in the Niño 3.4 region (https://www.esrl.noaa.gov/psd/gcos_wgsp/ Timeseries/Nino34/). The monthly PDO index is defined as the leading principal component of monthly SST anomalies in the North Pacific basin (Zhang et al., 1997). It is obtained from the Joint Institute for the Study of the Atmosphere and Ocean (http://research.jisao.washington.edu/pdo/). For NAO, we use the station-based seasonal NAO index, which is the difference in normalized sea level pressure between Lisbon and Reykjavik (Hurrell and Deser, 2010) (https://climatedataguide.ucar.edu/ climate-data/hurrell-north-atlantic-oscillation-nao-index-station-based). Finally, the AMO index is defined as area-weighted average SST over the North Atlantic basin (Enfield et al., 2001). We use the monthly de-trended and un-smoothed AMO index derived from the Kaplan SST dataset (https://www.esrl.noaa.gov/psd/gcos_wgsp/Timeseries/AMO). For the PDO and AMO indices, we calculate 3-month running means to maintain seasonal persistence.

Monthly soil moisture and snowfall data are obtained from the ERA-40 reanalysis, developed by the European Centre for Medium-Range Weather Forecasts (https://apps.ecmwf.int/datasets/) and WATCH forcing data, respectively. For soil moisture, we aggregate all four volumetric soil water layers of ERA-40. To properly account for the basin-scale soil moisture and snowfall states (Maurer and Lettenmaier, 2004), we calculate the area-weighted average soil moisture and snowfall of all upstream grids for each dam using the DDM30 river network.





# 3 Methods

The goal of this study is to (1) quantify the value of seasonal inflow forecasts for a global database of hydropower dams, (2)
illustrate how reservoir design properties and forecast skill affect the value of seasonal forecasts, and (3) identify regions that
could benefit from application of seasonal forecasts. To achieve these goals, we first develop an inflow prediction model for
each of the 753 dams (Section 3.1). Then, we simulate hydropower production for each dam under three operating schemes
that are based on perfect forecasts, realistic forecasts (issued by our inflow prediction model), and (no forecast) control rules
(Section 3.2). Finally, we evaluate the performance of each operating scheme and identify the reservoir design specifications
that influence each system's performance (Section 3.3).

## 3.1 Dam inflow prediction model

Two broad alternative approaches for seasonal streamflow forecast development include physically-based models, such as
GloFAS (a global-scale forecasting system; Emerton et al. (2018); Harrigan et al. (2020)), or statistical prediction models that
leverage the relationship between large-scale climate drivers and local hydro-meteorological processes (Block, 2011; Gelati
et al., 2014; Giuliani et al., 2019). Here, we select the second approach for two reasons. First, re-forecasts issued by global-
scale forecasting systems are only available for a relatively-short hindcast period (typically two decades; Harrigan et al. (2020)),
whereas the time series of globally-available hydro-climatological data are significantly longer. Second, the prediction horizon
of most physically-based approaches (a few days to 3-4 months) falls short of our preferred lead times up to seven months, so
as to test the potential of realistic forecasts for a broad spectrum of reservoirs—including those characterized by slow storage
dynamics.

Our long-range inflow prediction models uses Principal Component Regression (PCR) and includes four lagged large-scale
climate drivers and prior streamflow conditions to predict streamflow at 1,200 stations, following Lee et al. (2018). This
approach is readily implemented globally and has demonstrated fair (realistic) predictive skill (Lee et al., 2018). While Lee
et al. (2018) predict seasonal (3-month) streamflow averages, here we develop independent monthly prediction (MP) models
for the subsequent seven calendar months. For example, forecasts issues at the end of February include monthly inflows from
March (MP1) to September (MP7).

The methodology relies on the following steps, illustrated in Figure 1. First, we normalize (log-normalize for streamflow)
and detrend all predictors and streamflow observations to avoid spurious correlation. Then, we estimate the lag-correlations
between monthly inflows over the next 7 months and climate indices (1-8 months ahead), snowfall (current to 8 months ahead),
inflow and soil moisture (current month). Only statistically significant predictors are subsequently used to develop the MP
models. If only a single (statistically significant) predictor exists, we apply a linear regression (LR) model; otherwise, we apply
the PCR model to avoid possible multi-collinearities. In the PCR process, we truncate only the last principal component, which
is typically associated with multi-collinearities, as suggested by Jolliffe (2002) and Wilks (2011). To select the optimal lead-
times of the lagged predictors, we apply a leave-one-out cross-validation (LOOCV) scheme. Specifically, all combinations
of lead-months for the lagged predictors are cross-validated; then, the optimal set of lead-months is determined based on





the minimum mean squared error (MSE). The models are developed with 70% of the available data (corresponding to the period 1958–1987) and validated with the remaining data (1988–2000), so as to measure the model's independent performance over the recent period. In the validation process, we evaluate the model performance using two skill scores, namely the mean squared error skill score (MSESS) and the Gerrity skill score (GSS) (Text S2 and Figure S2). If an MP model has no statistically

significant predictors, or either an MSESS or GSS value less than 0, the long-term average for that month (i.e. climatological mean) is applied instead. The overall accuracy of the reservoir inflow predictions is assessed with the Kling-Gupta efficiency ($KGE$), which compares correlation, bias, and variability of the predicted and observed discharge (Gupta et al., 2009). The KGE is defined as:

$$KGE = 1 - \sqrt{(r-1)^2 + (\beta-1)^2 + (\gamma-1)^2},$$ (1)

where $r$ is the correlation coefficient, $\beta$ the bias ratio of the mean inflow ($\mu_s/\mu_o$), $\gamma$ the variability ratio ($CV_s/CV_o$), $\mu$ the mean flow, $CV$ the coefficient of variation, and $s$ and $o$ are two indices indicating simulated (predicted) and observed inflow values, respectively.

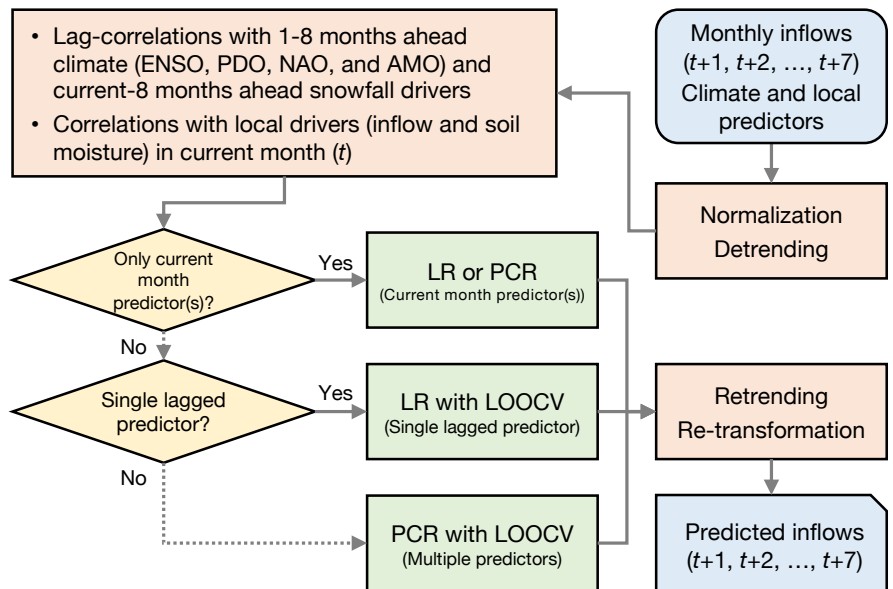

**Figure 1.** Graphical representation of the monthly prediction (MP) model scheme. At each calendar month $t$, we develop seven independent models to predict monthly inflows for the next seven months: MP1 ($t+1$), MP2 ($t+2$), ..., MP7 ($t+7$).



## 3.2 Reservoir operation model

### 3.2.1 Reservoir model

An essential component of the operating scheme is the reservoir mass balance described below:

$$S_{t+1} = S_t + Q_t - E_t - R_t - Spill_t, \tag{2a}$$

$$0 \leq S_t \leq S_{cap}, \tag{2b}$$

$$0 \leq R_t \leq \min(S_t + Q_t - E_t, R_{max}), \tag{2c}$$

where $S_t$ is the reservoir storage at month $t$, $Q_t$ the inflow volume (retrieved from the WaterGAP model, as described in Section
2.1), $E_t$ the evaporation loss, and $R_t$ the water released through the turbines. Both $S_t$ and $R_t$ are constrained by the reservoir design specifications. Specifically, the storage cannot exceed the reservoir capacity $S_{cap}$ (eq. (2b)), while the discharge is bounded by the water availability and capacity $R_{max}$ of the turbines (eq. (2c)). Excess water, if any, is spilled:

$$Spill_t = \max(0, S_t + Q_t - R_t - E_t - S_{cap}). \tag{2d}$$

The hydropower production $P_t$ (in MW) is calculated as follows:

$$P_t = \eta \cdot \rho \cdot g \cdot r_t \cdot h_t, \tag{3}$$

where $\eta$ is the efficiency of the turbines (assumed constant at 0.9 over the simulation period), $\rho$ the water density (1,000 kg/m$^3$), $g$ the gravitational acceleration (m/s$^2$), $r_t$ the average release rate (m$^3$/s) implied by the monthly release volume $R_t$, and $h_t$ the hydraulic head (m). The latter is taken as the average head between time $t$ and $t+1$.

### 3.2.2 Benchmark scheme: control rules

Our benchmark operating scheme relies on the approach proposed by Ng et al. (2017), in which the behaviour of the hydropower operators is modelled as an optimal control problem. This approach builds on two main assumptions. First, the goal of the operators is to maximize hydropower production over the long term. This objective provides a tangible indication of hydropower performance, so it is commonly adopted in large-scale studies (e.g., Van Vliet et al. (2016)). Second, the release decision $R_t$ depends on the reservoir storage $S_t$, the previous period's inflow volume $Q_{t-1}$, and month of year $t$—a common





choice in real-world reservoir operating schemes (Hejazi et al., 2008). In other words, the approach assumes that each reservoir
is operated through a unique periodic look-up table of turbine release decisions, which is generated through stochastic dynamic
programming (Loucks et al., 2005; Soncini-Sessa et al., 2007). In the optimization, the inflow process is modelled with a first
order, periodic Markov chain, whose parameterization is derived from the inflow data. This means that climatology and inter-
annual inflow variability are embedded in this operating scheme. Such schemes are more sophisticated than operating schemes

relying only on storage level and time of the year (Denaro et al., 2017; Giuliani et al., 2019; Ahmad and Hossain, 2020) and
are more consistent with inferred operating rules (Turner et al., 2020). A detailed validation of the operating rules—based on
values of observed hydropower production in 107 countries during the period 1980–2000—is reported in Turner et al. (2017b).
The time series of all process variables (e.g., inflow, storage, release, hydropower production) obtained by the benchmark
control scheme are available on HydroShare (http://www.hydroshare.org/resource/ca365ffb1a1f49df8b77e393be965fd8).

### 3.2.3    Forecast-informed scheme

To assess the value of seasonal streamflow forecasts, we adopt an adaptive scheme based on the *receding horizon principle*
(Bertsekas, 1976). At month $t$, we use a deterministic 7-month streamflow forecast to determine the value of the release
decisions for the next seven months, and then implement only the decision $R_t$ for the first month. At month $t + 1$, when a
new 7-month forecast becomes available, a new sequence of release decisions is determined. Each decision-making process

is formulated through an optimization problem that maximizes the hydropower production over the forecast horizon while
accounting for the benefits associated with the resulting storage at the end of the forecast horizon:

$$\min_{R_t, R_{t+1}, \ldots, R_{t+6}} \sum_{i=0}^{6} P_{t+i} + X(S_{t+7}), \tag{4}$$

where $P_t$ is the hydropower production (see eq. (3)) and $X(\cdot)$ a function accounting for the long-term effect of the release
decisions. Specifically, the function penalizes decisions that solely optimize energy production in the short term, risking water

availability depletion in the long term. Following a common practice in forecast-informed schemes (Soncini-Sessa et al.,
2007), we set $X(\cdot)$ equal to the benefit function obtained by the benchmark control rules, which contains information about the
expected long-term hydropower production for a given storage level. Thus, the real-time information provided by the forecasts
may alter decisions otherwise based solely on the benchmark scheme. As our inflow forecast model gives a deterministic 7-
month forecast, the optimization problem is solved at each time step using deterministic dynamic programming (Turner et al.,

2017a).

    The scheme is implemented using both 'perfect' and realistic forecasts. Both benchmark and forecast-informed schemes
are simulated over the period 1958–2000. During the simulation, all release decisions are constrained to satisfy downstream
environmental flow requirements, calculated using the variable monthly flow method (Pastor et al., 2014). All experiments are
carried out with the R package *reservoir* (Turner and Galelli, 2016).





### 3.2.4 Considering additional operating objectives and finer temporal scales

Of the 735 headwater dams in the database, 174 dams are also operated for flood control purposes. For these dams, we penalize spill to account for flood control and formulate the optimization objective as follows (in both benchmark and forecast-informed schemes):

$$\min_{R_t, R_{t+1}, \ldots, R_{t+6}} \sum_{i=0}^{6} \left( w_1 \cdot \frac{Spill_{t+i}}{p_{95}(Q)} + w_2 \cdot \left(1 - \frac{P_{t+i}}{P}\right) \right) + X(S_{t+7}), \tag{5}$$

where $w_1$ and $w_2$ are the weights associated with the flood control and hydropower objectives (set to 0.5 here), $p_{95}(Q)$ the 95[th] percentile of the inflow time series $Q$, and $P$ the dam installed hydropower capacity (in MW). Clearly, additional objectives may influence hydraulic head or the release trajectory, thereby affecting hydropower production (Zeng et al., 2017).

A second modification of the reservoir operation model concerns the monthly decision-making time step, which may not be suitable for reservoirs with small storage capacity relative to inflow (time-to-fill). We therefore identify a group of 94 reservoirs for which the time-to-fill is shorter than two months, and adopt for this group only a weekly time step. Since the inflow forecasts have a monthly resolution, we disaggregate each forecast into four values using the $k$-nearest neighbors algorithm (Nowak et al., 2010). Further details are reported in Text S3.

### 3.3 Reservoir performance evaluation

It is reasonable to hypothesize that that the value of seasonal streamflow forecasts—here measured in terms of hydropower production—depends not only on predictive skill, but also on reservoir characteristics. For example, a reservoir constrained by small turbine capacity may perform adequately utilizing control rules alone, as storage is sufficient to buffer inflow variability. We are thus interested in quantifying forecast value as well as understanding how value varies as a function of both skill and reservoir characteristics. This leads us to the following performance metrics.

### 3.3.1 Impact of design characteristics on perfect forecast-based operations

Initially excluding the effect of actual forecast skill, the following performance metric represents the expected improvement from perfect forecast-informed operations as compared to control rules-based operations:

$$I_{PF} = \frac{H_{PF} - H_{ctrl}}{H_{PF}} \times 100\%, \tag{6}$$

where $H_{PF}$ and $H_{ctrl}$ represent the total hydropower production (for the period 1958–2000) obtained with perfect forecast-informed operations and control rules, respectively. A value equal to zero indicates that the control rules are comparable to the (perfect) forecast-informed operations, whereas a positive value suggests that forecast-informed operations could be beneficial. Even though some dams are operated with an additional flood control objective, we use the same measure for all dams so as to ensure a consistent comparison of forecast value.





To understand how reservoir characteristics may influence benefits attained with perfect forecasts, we proceed in two steps. First, we label each dam as *success* or *failure* depending on whether the associated value of $I_{PF}$ is larger or smaller than
the mean value of $I_{PF}$ across all dams. (As this labelling is rather arbitrary, we later assess the sensitivity of our analysis to changes in the threshold.) Note that *failure* implies that the control rules and perfect forecast-informed operations generate a similar amount of hydropower, meaning that information on storage and previous-month inflow are sufficient for near-optimal release decisions. Second, we explain the likelihood of achieving success through a logistic regression model in which the probability of the binary response variable taking a particular value is a function of the predictor variables. We consider two
predictors, namely (1) the ratio of reservoir storage capacity to the mean monthly inflow ($x_{fill}$, measured in months), and (2) the ratio of maximum reservoir depth to maximum hydraulic head ($x_{depth}$). The second predictor varies between 0 and 1, and indicates the extent to which hydraulic head is dependent on the depth of the reservoir. The logistic regression model is cross-validated with a 10-fold cross-validation scheme, and evaluated using two metrics, accuracy and Cohen's kappa (McHugh, 2012). Accuracy is the ratio of correctly predicted observations (true positives and true negatives) to the total number of
observations. Cohen's kappa is an adjusted accuracy score that accounts for the possibility of correct predictions occurring by chance. The modelling exercise is carried out with the R package *caret*. For additional details, please refer to Text S4, and Table S1–S3 in the Supplement.

### 3.3.2 Impact of forecast skill and design characteristics on realistic forecast-based operations

Integrating realistic forecasts in lieu of perfect forecast information, we introduce the following performance metric:

$$I_{DF} = \frac{H_{DF} - H_{ctrl}}{H_{PF}} \times 100\%, \tag{7}$$

where $H_{DF}$ represents the total hydropower production (for the period 1958–2000) obtained using realistic forecast-informed operations. $I_{DF}$ is then combined with $I_{PF}$ to calculate the performance metric $I$ that quantifies the potential improvement between realistic and perfect forecast-informed operations:

$$I = \frac{H_{DF} - H_{ctrl}}{H_{PF} - H_{ctrl}} = \frac{I_{DF}}{I_{PF}}. \tag{8}$$

A value of $I$ equal to 1 indicates that benefits from the actual forecasts equal those utilizing perfect forecasts. A value of 0 denotes performance equivalent to applying the control rules only, while a negative value implies that the forecast-informed scheme is inferior to the control rules. We calculate this metric only for the subset of dams achieving a value of $I_{PF}$ greater than the mean value of $I_{PF}$ to better understand if the benefits modeled with perfect forecasts may be attainable with realistic forecasts.

To explain how the metric $I$ varies, we use a linear regression model accounting for both forecast skill and reservoir characteristics. The predictor characterizing the forecast skill is $x_{MdAPE}$, the median absolute percentage error of the forecast, used in place of $KGE$ because it shows a higher correlation with $I$. (While $KGE$ gives a broad view of the forecast skill by





comparing correlation, mean, and standard deviation of the predicted and observed inflows, $MdAPE$ accounts for the forecast error at every time step of the inflow time series. This may make $MdAPE$ a more suitable predictor, as the error at each time
step affects the release decisions and, ultimately, hydropower production.) The second predictor is $x_{exceed}$, the fraction of time that inflow exceeds the maximum turbine release rate. For more details on the choice of predictors, please refer to Text S5 and Table S3–S4.

## 4 Results

In this section, we first present the accuracy of the inflow prediction models (Section 4.1) and performance of the forecast-
informed schemes (Section 4.2). Then, we quantify the extent to which reservoir design characteristics and forecast skill affect the value of seasonal forecasts (Section 4.3). Lastly, we classify all dams according to their potential to benefit from forecasts, and identify key geographical regions that may benefit the most from forecasts (Section 4.4).

### 4.1 Potential predictors and accuracy

As shown in Figure 2, reservoir inflow exhibits significant correlation with climate and local drivers (potential predictors).
Yet, this relationship changes across the annual cycle. Evaluating months when a higher percentage of dams is significantly correlated with predictors, some well-known climatic teleconnections can be observed—e.g., ENSO and winter-spring stream-flow in North America and Europe, NAO and spring-summer peak flows in the northern extratropic regions, and PDO and summer streamflow in southeastern North America and central South America (Figure S3). On average, 27%, 37%, 28%, 20%, and 36% of the reservoir catchments are significantly correlated with ENSO, NAO, PDO, AMO, and snowfall, respectively.
Additionally, and not surprisingly, inflow for most dams (72%) exhibits significant 1-month lead autocorrelation. An exception is represented by some dams during the period March-April, especially in areas with minimal baseflow (Figures 2 and S3). Soil moisture at a 1-month lead is statistically significantly correlated with inflow at 47% of dams across all months with a seasonality similar to inflow.

Reservoir inflow and climatic predictors are often (significantly) correlated across several lead months. In these cases, climate
predictors are very likely to be included in multiple MP models for various leads, although the correlation may decrease with longer lead-time. When a climate predictor is significantly correlated with reservoir inflow at a 1-month lag (MP1), 74% and 38% of the time it is also included at the 4-month lag (MP4) and 7-month lag (MP7), respectively. Snowfall has a similar retention rate. However, and as expected, autocorrelation in inflow and soil moisture drops more precipitously with longer lead; only 53% (28%) of the time, when lagged inflow and soil moisture are included as predictors in MP1, they are also included in
MP4 (MP7). Globally, an average of 2.7, 1.7, and 0.9 predictors are included in the MP1, MP4, and MP7 models, respectively. In very few cases, the number of predictors increases with longer lead-time. For months when no potential predictors are identified, or either MSESS or GSS is less than zero, the long-term mean inflow for that month is used.

Across the annual cycle, the average number of months in which at least one predictor is included (and thus a predictive model developed) is equal to 8.3 months (MP1), 6 months (MP4), and 4.2 months (MP7) (see Figure 3). As noted previ-





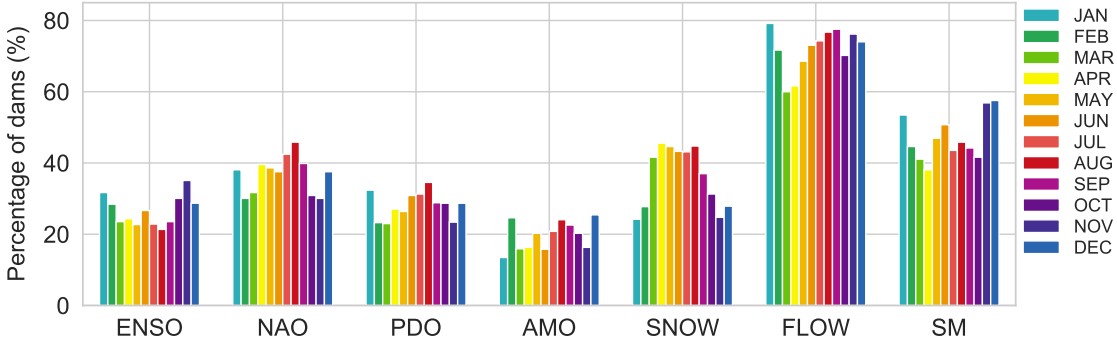

**Figure 2.** Percentage of dams significantly correlated with lagged predictors (ENSO, NAO, PDO, AMO, and snowfall) and 1-month ahead predictors (inflow and soil moisture) in each calendar month.

ously, a lack of long-lead inflow autocorrelation is predominantly responsible for this drop (and thus for increased reliance on climatology-based forecasts). Prediction accuracy also decreases with lead-time; average $KGE$ values are 0.64 and 0.56 for MP1 and MP7, respectively (Figure 3). Given that prediction accuracy generally declines with lead time, the highest $KGE$ scores across all MP models are associated with MP1 for 68% of the dams. For the remaining models, the highest prediction accuracy is recorded for 5% (2%) of dams in the MP4 (MP7) models, emphasizing that skillful forecasts at longer leads do

exist, such as in Europe or northwestern and southeastern U.S. (Figure 3b and 3c). As for the geographical distribution of $KGE$, we find relatively high $KGE$ scores in several regions, including North America, eastern South America, Europe, and some regions in western Africa and Asia, where inflows correlate with most of the considered predictors (Figures 3 and S3).

For all MP models, the $KGE$ has an average value of about 0.56, which is regarded as a fair skill score outcome. While uniquely tailored forecasts could be produced for each dam considering more local influences, the current prediction approach

performs well globally and reflects achievable long-range inflow predictions. Considering the superior performance of the MP1 model, the forecast skill of MP1 only is retained to represent the overall forecast skill in the following analyses.

## 4.2 Performance of forecast-informed operations

The expected performance of perfect and realistic forecast-informed operations is notably different across the 735 hydropower dams (Figure 4). With perfect forecast-informed operations (Figure 4a), we observe a substantial increase in hydropower

production with respect to the baseline control rules. Specifically, 94% of dams exhibit a positive value of the performance metric $I_{PF}$; mean improvement is 4.7% and maximum improvement is 60%. For the small number of dams that do not benefit from perfect forecasts, the value of $I_{PF}$ does not drop below -1.7%. Small negative values of $I_{PF}$ are likely a result of the discretization needed by dynamic programming to optimize the release sequence (eq. (4)), hence allowing control rules to outperform perfect forecast-informed operations. Considering all dams collectively, an additional 24 TWh per year

of hydroelectricity are generated when adopting the perfect forecast-informed approach in lieu of baseline control rules. This is equivalent to 0.57% of the 4,200 TWh of hydropower globally generated in 2018 (note that the headwater dams used here



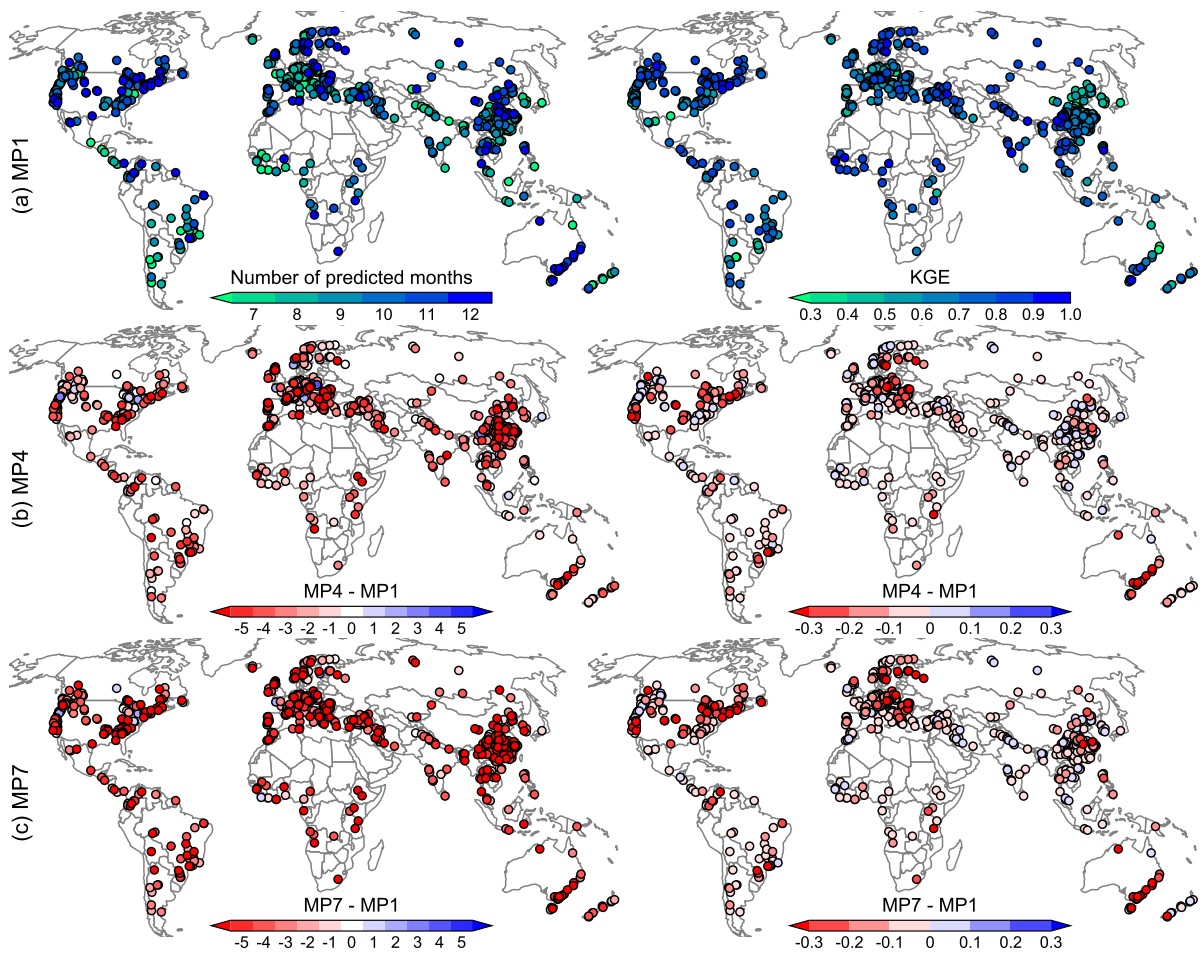

**Figure 3.** Number of months in which a predictive model is developed (left) and corresponding $KGE$ (right). Taking a model with a lead-time of 1 month (MP1) as reference (a), we report the difference between MP1 and MP4 (b) and MP1 and MP7 (c).

represent 10% of the world's installed capacity) (IHA, 2019). Such modest global benefit and large range of individual benefits suggest that the forecast value is highly dependent on the reservoir characteristics (see Section 4.3).

When realistic forecast-informed operations are adopted (Figure 4b), a smaller number of dams exhibit increased hy-
dropower production. $I_{DF}$ ranges from -24% to 28%, with 25% of dams showing a positive value of $I_{DF}$. The 184 dams with positive $I_{DF}$ values show an average improvement of 2.3% and collectively contribute an additional 1.7 TWh per year in hydropower production—7% of the 24 TWh of additional hydropower obtainable from perfect forecasts. This decline in performance is expected, as realistic forecasts introduce a non-negligible prediction error. Yet, it should also be noted that less than 20% of dams have a $KGE$ value below 0.5, whereas a disproportionately larger number of dams exhibit a negative $I_{DF}$
value. This suggests that for a large number of dams control rule-based operations are superior to realistic forecast-informed



**Figure 4.** Improvements in hydropower production using perfect (a) and realistic (b) forecasts. The terms $I_{PF}$ and $I_{DF}$ indicate the relative improvement in hydropower production (with respect to the basic control rules) provided by perfect and realistic forecasts. Nearly all dams are able to benefit from perfect forecasts, but only 25% of dams benefits from realistic forecasts.

operations. For dams with poor $I_{DF}$ and high $KGE$, two features are noteworthy: first, $KGE$ may not fully capture the relationship between forecast skill and value; and, second, reservoir characteristics are an important factor influencing the value of realistic forecasts.





### 4.3 Evaluation of prediction accuracy and reservoir characteristics

To understand the extent to which reservoir characteristics may modulate the value of seasonal forecasts, we identify a logistic regression model that explains the likelihood of achieving success with perfect forecasts (i.e., $I_{PF}$ larger than 4.7%, the mean value of $I_{PF}$ across all dams) as a function of two predictors, $x_{fill}$ (the ratio of reservoir storage capacity to the mean monthly inflow) and $x_{depth}$ (the ratio of maximum reservoir depth to maximum hydraulic head). A 10-fold cross-validation yields a model accuracy and Kappa statistic of 0.785 and 0.535. (Note that the percentage of dams falling into the *success* and *failure*
categories is equal to 37% and 63% respectively.)

As illustrated in Figure 5 and Table 1, both predictors influence the probability of achieving success. For $x_{fill}$ values exceeding ten months, dams are highly unlikely to benefit substantially from seasonal forecasts. This suggests that a large storage capacity effectively acts as a buffer against inflow uncertainty. Hence, both control rules and perfect forecast-informed operations tend to attain similar performance. We also observe that some of the smaller dams ($x_{fill}$ <2) fail to attain increased
hydropower production even though they are predicted to do so (red triangles in the blue shaded region; Figure 5). This is attributed to the weekly operations, suggesting that more frequent release decisions may reduce forecast value, since the benchmark operating rules have more opportunities to adjust release decisions. For smaller dams, $x_{depth}$ becomes a critical factor. High values of $x_{depth}$ indicate that the hydraulic head is highly dependent on the reservoir depth, which is in turn dependent on current and near future inflows for dams that cannot accumulate large inflow volumes. Thus, forecast-informed
operations become crucial to maintain a high hydraulic head and maximize hydropower production. For hydropower dams that have a low value of $x_{depth}$, a high hydraulic head is maintained even when storage is low, thereby minimizing the utility of forecasts. These are systems relying on waterfalls, or hilly terrains, to divert part of the water and gain hydraulic head.

**Table 1.** Coefficients of logistic regression to predict if $I_{PF} > 4.7\%$ . The term 'Estimate' represents the increase in log-odds of a dam attaining *success* per unit increase in the value of the predictors.

| Predictors | Estimate | Std. Error | Z-value | Pr(>\|z\|) |
|---|---|---|---|---|
| (Intercept) | -1.16 | 0.25 | -4.62 | <0.01 |
| $x_{depth}$ | 2.84 | 0.31 | 9.25 | <0.01 |
| $x_{fill}$ | -0.10 | 0.01 | -8.76 | <0.01 |

Considering only the subset of 269 dams that have an $I_{PF}$ value larger than 4.7%, we apply a linear regression model to estimate the performance metric $I$. This time, the predictors include $x_{MdAPE}$ (median absolute percentage error of forecast
inflows) and $x_{exceed}$ (the fraction of time that inflow exceeds the maximum turbine release rate). The linear regression model has an adjusted $R^2$ of 0.31, which can be increased further by considering other variables related to inflow variability and hydraulic head. The reader is referred to Table S4-5 for more complex models that include additional predictors.

The results are presented in Table 2 and illustrated in Figure 6. As expected, higher forecast skill (lower $x_{MdAPE}$) increases the potential benefits realized by the realistic forecasts; a 1% decrease in $x_{MdAPE}$ increases $I$ by 0.03. Reservoir characteristics
can play an important role, as certain configurations allow dams and hydropower production to benefit from realistic forecasts.




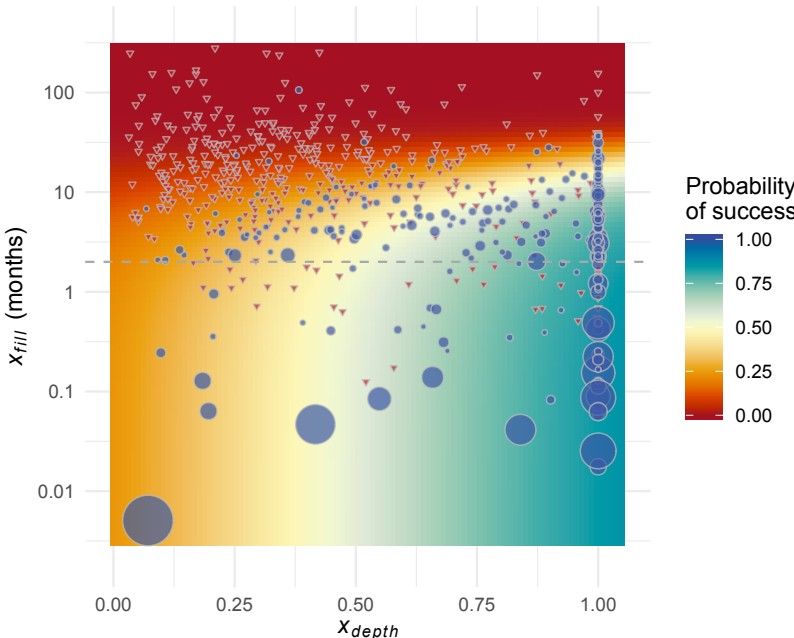

**Figure 5.** Probability of *success* estimated using a logistic regression model with predictors $x_{depth}$ and $x_{fill}$ (in log scale). Red corresponds to a probability of success equal to zero, meaning that the dam is likely to do well with the control rules. Blue represents a probability of success equal to 1, meaning that a dam is likely to benefit from forecast-informed operations. Each point in the plot represents one of the 735 dams. Blue circles represent dams labelled as *success* ($I_{PF} > 4.7\%$) and red triangles represents *failures*. The size of the blue circles represents the value of $I_{PF}$. All red triangles have the same size. Dams below the dashed line ($x_{fill} = 2$) are operated with a weekly time step. Dams with low values of $x_{fill}$ (small storage capacity relative to inflow rate) and high $x_{depth}$ (lacking a natural waterfall) are more likely to benefit from forecast-informed operations.

Specifically, we find that dams in which inflow frequently exceeds maximum turbine release (large values of $x_{exceed}$) are more likely to benefit from forecast-informed operations—even when forecasts are not very accurate, as shown by the diagonal divide in Figure 6. This is predominantly a result of both forecast and observed inflow frequently exceeding the maximum turbine release rate, a situation in which the release decision would be the same regardless, so consequently inaccurate forecasts do not penalize hydropower production.

**Table 2.** Coefficients of linear regression to predict *I*.

| Predictors | Estimate | Std. Error | Z-value | Pr(>\|z\|) |
|---|---|---|---|---|
| (Intercept) | -0.18 | 0.12 | -1.485 | 0.139 |
| $x_{MdAPE}$ | -0.03 | 0.004 | -8.554 | <0.01 |
| $x_{exceed}$ | 2.36 | 0.30 | 7.752 | <0.01 |



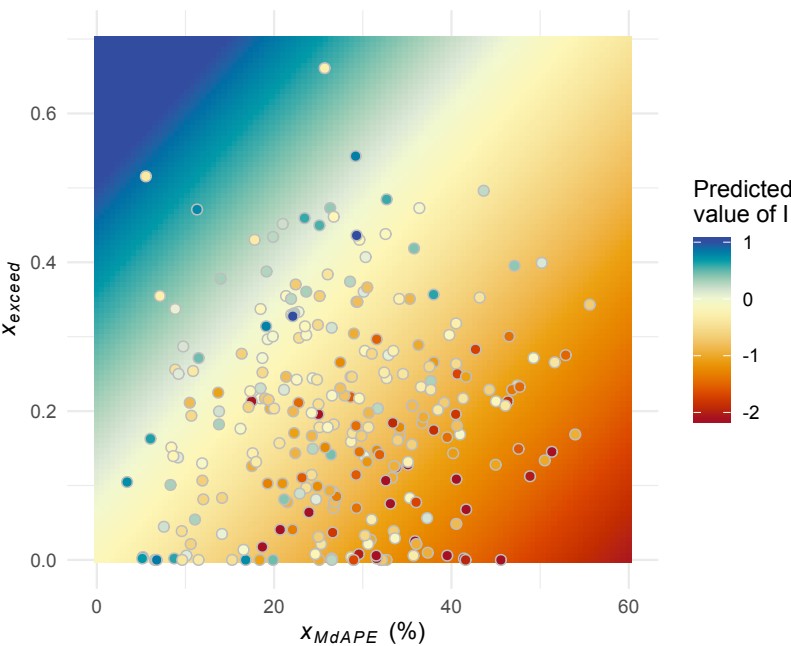

**Figure 6.** Potential benefits realized by realistic forecast ($I$) predicted using linear regression with predictors $x_{exceed}$ and the median absolute percentage error ($x_{MdAPE}$). Red corresponds to negative values of $I$, meaning that the performance of realistic forecasts is worse that the one attained by control rules. Blue corresponds to positive values of $I$, meaning that realistic forecasts outperform control rules. Each point corresponds to one of the 269 dams with $I_{PF} > 4.7\%$. The corresponding color represents the value of $I$ attained via simulation with the reservoir operation model. Dams with accurate forecasts and high values of $x_{exceed}$ (inflow frequently exceeds maximum turbine release) tend to have greater hydropower benefits realized from realistic forecasts.

## 4.4 A classification of hydropower dams

Building on the results described above, we divide the dams into four groups on the basis of their potential to benefit from perfect forecast-informed operations (*high potential* if $I_{PF} > 4.7\%$ and *low potential* otherwise) and forecast skill (*good forecast* if $x_{MdAPE} < 20\%$ and *poor forecast* otherwise). The cut-off value for $I_{PF}$ is inferred from the previous analysis (logistic regression model), while the cut-off for $x_{MdAPE}$ divides the 735 dams into two groups of one third (*good forecast*) and two thirds (*poor forecast*) of the observations. The distribution of the dams across the four groups (and Köppen-Geiger climate zones) is reported in Table 3. Two groups of dams of particular interest include (1) dams that fall in regions expressing strong forecast accuracy and have the potential to reap benefits from forecast-informed operations (9% of the total number of reservoirs), and (2) dams with strong potential to benefit from forecast-informed operations but lack forecast accuracy (28%). As discussed in Section 5, this lack of forecast accuracy could be readily addressed by adopting more sophisticated forecast models or further leveraging local predictors.




**Table 3.** Distribution of dams across climate zones. In columns 2-9, H and L indicate the potential (*high/low*) of benefiting of forecasts, while G and P indicate the quality (*good/poor*) of realistic forecast. Columns 2-5 (6-9) are the number (percentages) of dams in each group. The last two columns report the percentages of dams with *high potential* and *good forecast*, respectively. Values reported in bold indicate whether the observed frequency is statistically different from the expected frequency (global average in the final row) ($p < 0.05$ using $\chi^2$ test).

| Climate | HG | HP | LG | LP | HG% | HP% | LG% | LP% | High% | Good% |
|---|---|---|---|---|---|---|---|---|---|---|
| Af | 0 | 9 | 3 | 3 | 0.00 | 0.60 | 0.20 | 0.20 | 0.60 | 0.20 |
| Am | 4 | 6 | 5 | 4 | 0.21 | 0.32 | 0.26 | 0.21 | 0.53 | 0.47 |
| Aw | 7 | 6 | 26 | 4 | 0.16 | 0.14 | 0.61 | 0.09 | 0.30 | **0.77** |
| BWh | 2 | 2 | 2 | 0 | 0.33 | 0.33 | 0.33 | 0.00 | 0.67 | 0.67 |
| BWk | 0 | 3 | 1 | 0 | 0.00 | 0.75 | 0.25 | 0.00 | 0.75 | 0.25 |
| BSh | 1 | 2 | 4 | 6 | 0.08 | 0.15 | 0.31 | 0.46 | 0.23 | 0.39 |
| BSk | 0 | 2 | 6 | 5 | 0.00 | 0.15 | 0.46 | 0.39 | 0.15 | 0.46 |
| Csa | 4 | 11 | 9 | 19 | 0.09 | 0.26 | 0.21 | 0.44 | 0.35 | 0.30 |
| Csb | 6 | 15 | 4 | 10 | 0.17 | 0.43 | 0.11 | 0.29 | **0.60** | 0.29 |
| Cwa | 5 | 25 | 8 | 16 | 0.09 | 0.46 | 0.15 | 0.30 | **0.56** | 0.24 |
| Cwb | 1 | 3 | 7 | 2 | 0.08 | 0.23 | 0.54 | 0.15 | 0.31 | 0.62 |
| Cfa | 2 | 56 | 6 | 58 | 0.02 | 0.46 | 0.05 | 0.48 | **0.48** | **0.07** |
| Cfb | 9 | 15 | 9 | 41 | 0.12 | 0.20 | 0.12 | 0.55 | 0.33 | 0.24 |
| Dsa | 0 | 2 | 1 | 2 | 0.00 | 0.40 | 0.20 | 0.40 | 0.40 | 0.20 |
| Dsb | 2 | 3 | 5 | 4 | 0.14 | 0.21 | 0.36 | 0.29 | 0.36 | 0.50 |
| Dsc | 0 | 1 | 1 | 1 | 0.00 | 0.33 | 0.33 | 0.33 | 0.33 | 0.33 |
| Dwa | 0 | 5 | 1 | 8 | 0.00 | 0.36 | 0.07 | 0.57 | 0.36 | 0.07 |
| Dwb | 1 | 1 | 2 | 1 | 0.20 | 0.20 | 0.40 | 0.20 | 0.40 | 0.60 |
| Dwc | 2 | 1 | 2 | 0 | 0.40 | 0.20 | 0.40 | 0.00 | 0.60 | 0.80 |
| Dfa | 0 | 1 | 6 | 5 | 0.00 | 0.08 | 0.50 | 0.42 | 0.08 | 0.50 |
| Dfb | 13 | 21 | 32 | 41 | 0.12 | 0.20 | 0.30 | 0.38 | 0.32 | 0.42 |
| Dfc | 7 | 10 | 33 | 20 | 0.10 | 0.14 | 0.47 | 0.29 | **0.24** | **0.57** |
| ET | 1 | 2 | 7 | 36 | 0.02 | 0.04 | 0.15 | 0.78 | **0.07** | **0.17** |
| Total | 67 | 202 | 180 | 286 | 0.09 | 0.28 | 0.25 | 0.39 | 0.37 | 0.34 |

As described in Section 4.3, the potential of a dam to benefit from forecasts is largely dependent on its design specifications, which present comparable values in areas with similar orography and design practices. Forecast skill, on the other hand, is largely dependent on climate teleconnections, which tend to present regional patterns. Further, considering these factors coincidentally is also insightful. Figure 7 illustrates the distribution of the four possible groups of dams across the thirty climate zones of the Köppen-Geiger climate classification system. We notice a few interesting patterns. First, dams with high potential but lacking accurate forecasts (panel (a), red triangles) are often found in humid subtropical climate zones ($Cwa$, $Cfa$), particularly in the southeast regions of Australia, China, U.S., and South America. The trend is also true—but not statistically significant—for dams in Southeast Asia (tropical rainforest, $Af$) and Pacific Northwest (warm summer Mediterranean, $Csb$). Second, dams with good forecasts (panel (a) and (b), blue triangles) are mostly located in the tropical savanna climate zone ($Aw$, Mainland Southeast Asia, India, Brazil, and western Africa) and subartic climate zone ($Dfc$, Canada, Russia, north eastern Europe). While the majority of these dams have poor to fair potential, which can be attributed to relatively large values of time-to-fill, the remaining dams with characteristics conducive for forecast-informed operations can readily benefit from



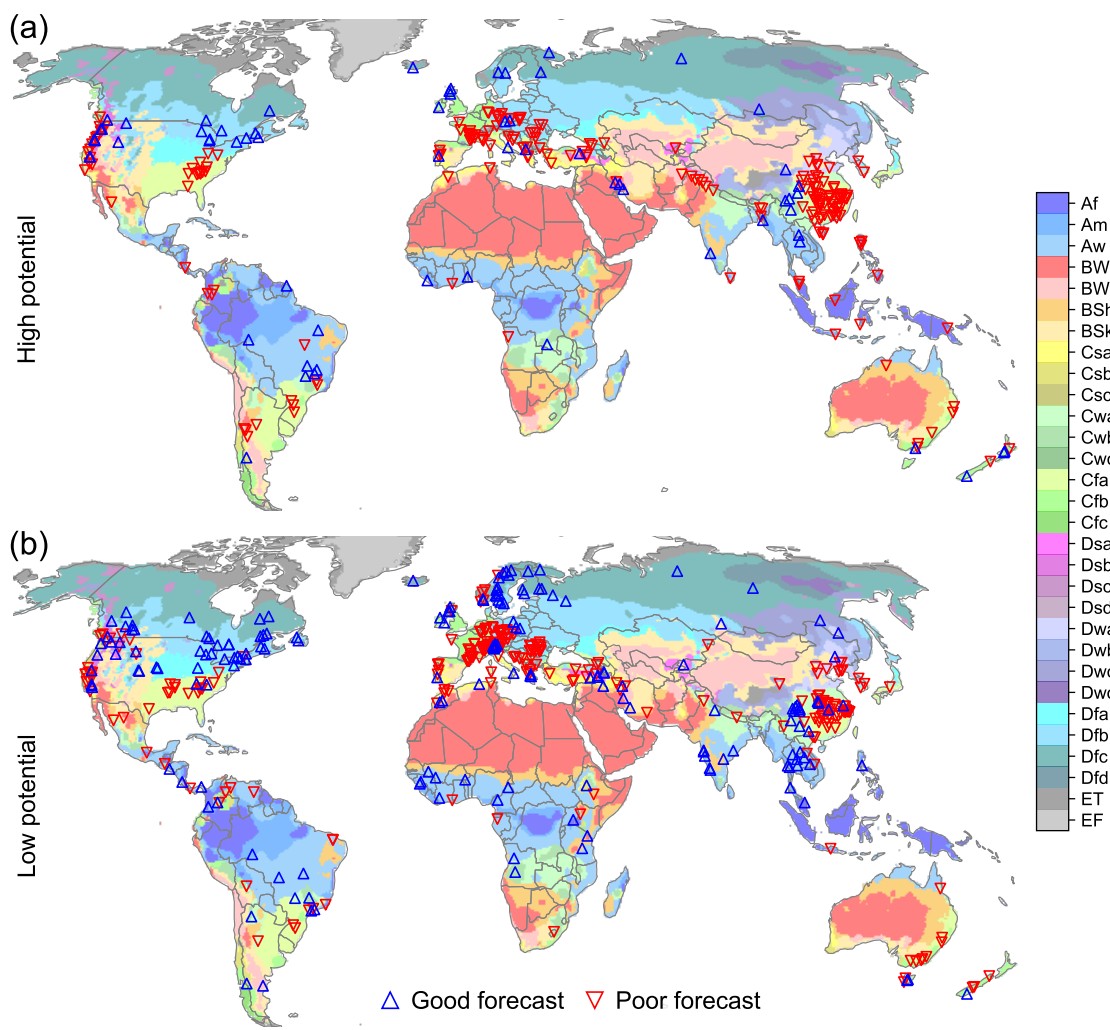

**Figure 7.** Distribution of dams across climate zones based on their potential to benefit from perfects forecasts. The top (a) and bottom (b) panels represent dams with 'high potential' ($I_{PF} > 4.7\%$) and 'low potential' ($I_{PF} \leq 4.7\%$), respectively, while 'good forecast' and 'poor forecast' represent dams with $MdAPE$ less than or greater than 20%, respectively.

including climate signals into their forecast models. This also applies to dams along the U.S.–Canadian border. These dams are located in a humid continental climate zone ($Dfb$), whose statistical significance for high forecast accuracy has been masked by differences in performance between dams located in Europe and North America. Third, dams with low potential and poor forecasts are primarily found in Europe and east Asia (panel (b), red triangles). This trend is significant ($p < 0.05$) for dams in the Alps (alpine climate, $ET$), in particular for dams that are characterized by large time-to-fill values and small reservoir depth to maximum hydraulic head ratios.





## 5   Discussion

### 5.1   Implications for planning and management of hydropower projects

In this study, we examine the relationship between seasonal streamflow forecasts and global hydropower production, accounting for the influence of reservoir characteristics. Specifically, we develop seasonal inflow forecasts for 735 headwater dams based on lagged global and local hydro-climatic variables. The forecasts exhibit fair skill globally, but higher skill in several regions, including the northern extra-tropical regions and the areas characterized by tropical savannah climate (e.g., mainland Southeast Asia, eastern South America, and western Africa). In agreement with earlier work, our forecasts exhibit well-known teleconnections, such as NAO influencing spring-summer peak flow in the northern extra-tropical regions (Lee et al., 2018), ENSO influencing streamflow in Southeast Asia (Sankarasubramanian et al., 2009; Räsänen and Kummu, 2013), and ENSO / PDO influencing winter-spring streamflow in the Pacific Northwest (Hamlet et al., 2002; Voisin et al., 2006).

We then illustrate the relationship between forecast skill, value, and reservoir characteristics by adopting forecasts in the reservoir operations model. While 94% of dams considered could benefit from perfect forecasts, only 25% demonstrate improvements when using our realistic forecasts—a fairly low percentage if we consider the forecast skill achieved globally. This highlights the fundamental role of reservoir characteristics in shaping the relationship between forecast skill and value. Key design specifications include time-to-fill, a characteristic identified by other recent studies (Anghileri et al., 2016; Turner et al., 2017a; Yang et al., 2020), hydraulic head, largely dependent on reservoir depth, and the frequency of inflows exceeding maximum turbine release rates, a design specification that allows operators to work with a larger margin of forecast error during high inflow periods. It is worth emphasizing here that these results are not intended to provide site-specific operational guidelines, but do represent a first, qualitative step toward determining the potential benefit of seasonal streamflow forecasts for hydropower operators. The relationships identified here could be used, for example, to understand forecast potential for a given reservoir or to characterize the interplay between climatology, hydrology, and dam characteristics in a large region of interest.

By combining information on reservoir characteristics, forecast skill, and climatic zones, we identify large regions in which dams would benefit the most from forecast application. One such group consists of dams with a strong potential to benefit from forecast-informed operations and that possess good forecast accuracy. In particular, for the tropical savanna and subartic climate, we observe teleconnections that result in higher forecast accuracy. Dams with favorable characteristics in these regions can gain additional benefits from integrating large-scale climate signals within forecast models. Another interesting group consists of dams that may benefit from forecast-informed operations but lack adequate forecast accuracy. Such dams are located in maritime Southeast Asia, the Pacific Northwest, and the humid subtropical climate of the southeast regions of Australia, China, U.S., and South America. These are areas in which watershed-specific analyses may bring immediate benefits to hydropower operators. Such analyses will likely require more nuanced hydro-climatological data than those adopted here—from observed precipitation in the upstream catchment (Denaro et al., 2017) to temperature and precipitation data forecasted by numerical weather prediction models (Ahmad and Hossain, 2020).





Finally, it is worth noting that the implications of our study go beyond existing reservoirs: dam planning over large scales may also benefit from these findings. For example, untapped hydropower potential (Zhou et al., 2015; Hoes et al., 2017)

and seasonal streamflow predictability could be evaluated to derive some first, qualitative, conclusions on expected reservoir characteristics and performance. A case in point are run-of-the-river dams: these systems have a short time-to-fill characteristic and are therefore suitable for implementing forecast-informed reservoir sizing and operations (Bertoni et al., 2021). Conversely, if new dams are constructed in areas known to lack forecast skill or monitoring systems, then a larger storage capacity may be justifiable for dams operating with basic control rules.

## 5.2 Limitations and opportunities

Like any other global study, the large spatial domain requires building on a number of assumptions that must properly contextualized. First, we assume that the goal of dam operators is to maximize hydropower production over the long term (in addition to providing flood control). While this objective provides a tangible indication of forecast value, it may not be fully representative of the local conditions encountered by operators. For example, operators may be interested to maximize revenue

(Anghileri et al., 2018), supply the bulk of power to the grid (Zambon et al., 2012), or complement the generation of other renewable energy sources (Graabak et al., 2019). To account explicitly for these aspects, one needs to model the role that dams play in the power market, as recently done for the Western U.S. (Voisin et al., 2020), England (Byers et al., 2020), or the Greater Mekong (Chowdhury et al., 2020, 2021). With these models, one could also infer the willingness to pay for improved streamflow forecasts based on the economic value derived from their use (Arnal et al., 2016).

Second, release decisions at individual dams may be affected by joint operations between multiple reservoirs and thus better supported with more accurate data and tailored hydrological models surpassing those adopted here. Importantly, these data could include qualitative or quantitative forecasts. Although precipitation and streamflow predictions are not used consistently across the world (Adams and Pagano, 2016), medium- to long-range forecasts are increasingly being adopted by water utilities—as recently shown by Turner et al. (2020) for 300 dams in the conterminous United States. Access to observed and

inferred release decisions could thus help researchers provide a more robust and nuanced estimate of forecast value.

Finally, investigation of alternative forecast approaches may be warranted. The adoption of a statistical prediction model is motivated by the availability of relatively long hindcast periods and the desire for long prediction horizons (Section 3.1). However, it is important to note that regional- and global-scale forecasting systems are gaining momentum (Kirtman et al., 2014; Emerton et al., 2018), showing skillful forecasts over large spatial domains and long prediction horizons (Arnal et al.,

2018; Towner et al., 2019), and may justify investigation. Importantly, such approaches may not be limited to headwater dams, as long as the underpinning hydrological model (coupled with seasonal forecasts) includes a realistic representation of water management decisions (Pechlivanidis et al., 2020).



# 6 Conclusions

This analysis expands the existing body of knowledge on the relationship between forecast skill, value, and reservoir design for the hydropower sector. As expected, a positive relation between skill and value exists, however we also demonstrate that value is strongly modulated by reservoir characteristics. The two extreme cases are represented by dams that can be profitable with little regard to forecast accuracy and dams that do not appear to benefit from seasonal streamflow forecasts. Considering reservoir characteristics and forecast skill together, we identify regions with high potential to benefit from forecast-informed operations whether forecast accuracy is good or poor. Research that integrates these findings with hydrological-electricity models to quantify economic benefits is warranted. Specifically, this may reflect the willingness to pay for improved forecast models. Such an assessment could provide guidance and insight for large-scale hydropower planning and management, particularly as energy systems become more interconnected.

*Code and data availability.* The code used to conduct all analyses is available by contacting the authors. All simulations results are available on HydroShare at http://www.hydroshare.org/resource/ca365ffb1a1f49df8b77e393be965fd8.

*Author contributions.* D.L., J.Y.N., S.G., and P.B. contributed to the conceptualization of this work. The data processing, analyses, and visualization were carried out by D.L. and J.Y.N. The first draft was prepared by D.L., J.Y.N., and S.G. All authors reviewed and edited the final draft.

*Competing interests.* The authors declare that they have no conflict of interest.

*Acknowledgements.* Jia Yi Ng and Stefano Galelli are supported by Singapore's Ministry of Education (MoE) through the Tier 2 project 'Linking water availability to hydropower supply—an engineering systems approach' (Award No. MOE2017-T2-1-143).



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
