# Peer review of "Unfolding the relationship between seasonal forecast skill and value in hydropower production: A global analysis"

_Hydrology and Earth System Sciences, 2021_

## Author Response (AR1)

Reply to reviewers of paper hess-2021-518

**Unfolding the relationship between seasonal forecast skill and value in hydropower production: A global analysis**

Donghoon Lee, Jia Yi Ng, Stefano Galelli, and Paul Block

Resilient Water Systems Group
Pillar of Engineering Systems and Design
SUTD, 8 Somapah Road
Singapore 487372
T. +65 6303 6600
http://people.sutd.edu.sg/~stefano_galelli/

**Editor**

Comments to the authors:

—---------------------------------------------------

Dear authors, I would like to thank you for your interest in HESS. I have carefully read the manuscript and find it well written, and would consider it to be an interesting contribution that fits within the scope of HESS. I would therefore propose to accept this for the discussion phase. There are some aspects that I am sure will be addressed by the reviewers. One comment I had to the geographic regions where the statistical model developed by the authors exhibited skill, is that while some regions that would be expected to have reasonable predictability indeed showed this, others did not; notably North-Western South America, the Horn of Africa, and Eastern Southern Africa. This may be due to the small sample size of dams in these regions, but it may warrant some additional comment.

—---------------------------------------------------

Dear authors, thanks again for the submission of your manuscript to HESS, which received very positive comments from the two appointed reviewers. The reviewers have, however, suggested some minor changes and clarifications. I would like to recommend that you consider these in revising the manuscript, and am looking forward to that revision.

—---------------------------------------------------

We would like to thank the Editor for the positive response and opportunity to revise our work. Following the reviewers' suggestions, we:

- Clarified a few important aspects related to our inflow prediction model (Section 3.1), as recommended by reviewer #1 and reviewer #2;
- Expanded the discussion (Section 5.2) to account for a few points mentioned by the reviewers. In particular, we elaborated on the uncertainties in streamflow forecasting;
- Better framed our inflow prediction model–and associated assumptions–in the current scene of streamflow forecasting (Section 3.1 and 5.2);
- Expanded the analysis to account for (1) the potential relationship between KGE and forecast value, and (2) an alternative classification of dams based on the hydro-climate classification introduced by Knoben et al. (2018) (instead of the Köppen-Geiger climate classification). Both points (raised by reviewer #1) did not lead to a major change in our results and conclusions, so we preferred to leave these analyses in the Supplement.
- Carried out a qualitative comparison between our forecast skill and the one achieved by the Global Flood Awareness System. In our response to reviewer #2, we also explain why we preferred to confine such comparison to our response-to-reviewers.

As for your comment concerning the regions that do not demonstrate effective forecast skill (e.g., Horn of Africa), we agree that this is not necessarily expected based on known teleconnections with precipitation; however, as you have noted, the limited number of dams

included in this study (unfortunately) contributes to the relatively low predictive skill presented. To clarify this point, we have added the following sentence at Line 347-349:

*For some regions that often exhibit skillful precipitation predictability based on well-known teleconnections and hydroclimate mechanisms, such as Northwestern South America, East Africa, and eastern Southern Africa (Lee et al., 2018), this is not readily apparent on the KGE maps (Figure 3), owing primarily to the regions' low dam density, and thus a small sample size.*

Finally, please note that our replies are highlighted in blue, while the revised sentences reported here are highlighted in blue and italics. Line numbers refer to the track-and-changes version of the manuscript.

**Reply to RC1: 'Comment on hess-2021-518', Anonymous Referee #1, 12 Jan 2022**

**Summary**

The manuscript by Lee et al. explores the relationship between forecast skill and value in the case of the management of hydropower dams. The authors use dam characteristics and forecast skill to identify categories of dams that (1) show potential for improvement or not over climatology-based operating rules, and (2) show improvement or not based on realistic forecasts. A climate classification is further used to "regionalize" the added value of long-term forecasts for the hydropower sector and identify regions where improvements of currently low quality forecasts would translate into added value for dam management.

The paper is of very high quality, is well referenced, well written and scientifically sound. It will be undeniably valuable for the forecasting community, but also has potential to reach hydropower production managers. Along with the manuscript come supplemental materials that further detail the methodology and the results, as well as a dataset and an R script that allow readers to access the datasets for each dam.

For these reasons, I strongly recommend this manuscript for publication. Hereafter, I list some questions to the authors, some recommendations for improving explanations, and mostly minor points.

We thank the reviewer for the positive comments and further critical comments that we believe have enhanced the overall quality of the manuscript.

**General comments**

Sections 2.1 and 4.4: You decided to use the Köppen-Geiger climate classification. Since you are working with hydropower and inflows which are influenced not only by climate patterns but also hydrological ones, a classification based on hydro-climate characteristics and not only climate characteristics would seem more relevant for the goal you are trying to achieve. Please consider using the hydro-climate classification proposed by Knoben et al. (2018).

Knoben, W. J. M., Woods, R. A., and Freer, J. E.: A Quantitative Hydrological Climate Classification Evaluated With Independent Streamflow Data, Water Resources Research, 54, 5088–5109, https://doi.org/10.1029/2018WR022913, 2018.

Thanks for suggesting to use the Hydrological Climate Classification (HCC) (Knoben et al., 2018). We looked into the key features of the HCC and identified two important points. First, the HCC is derived from climate variables, such as precipitation, temperature, and potential evapotranspiration (CRU TS v3.23), and then is evaluated with independent streamflow data. Therefore, the HCC could still be seen as a "climate-based" classification, although Knoben et al. (2018) showed that the HCC better represents streamflow characteristics in terms of grouping catchments. Second, the HCC is not a categorized classification like the Köppen-Geiger climate, but rather a set of three-dimensional numerical climate indices, that is, aridity, moisture seasonality, and snow fraction. As a result, categorizing dams according to their HCC is not a

straightforward process. Moreover, carrying out an analysis based on the HCC would go beyond our original intent, which was to complement the analysis of the four groups of dams we created.

This said, we agree that the HCC might reveal some additional insights, so we analyzed the relationships between HCC indices and forecast skills of 735 dams (see the figure below). For this analysis, we used averaged HCC values in the grids upstream of each dam. The analysis reveals a few interesting patterns: for example, dams in snowing regions (snow ≥ 0.2) tend to have good forecasts when seasonality is larger than 0.4 (panel b) or aridity is below 0.4 (panel c). Because of the reasons outlined in the previous paragraph we preferred to include the analysis in the supplementary information (please refer to Text S7 and Figure S5).

[Figure]

*Figure 1.1. Scatterplots contrasting values of the Hydrological Climate Classification indices (Knoben et al., 2018) for 735 dams: (a) seasonality vs aridity, (b) snow vs seasonality, and (c) aridity vs snow. Blue up-pointing (red down-pointing) triangles represent dams with good (poor) forecast skill based on $X_{MdAPE}$ cutoff value.*

Section 3.1, Figure 1: It is not clear to me why the authors allow future inflows (t+1, t+2, ...t+7) to be predicted based on future climate indicators (1-8 months ahead). In a true forecasting setting, the ENSO, PDO, NAO and AMO indices for the 1-8 months ahead would not be available, only forecasts of these indices would. Some clarifications would be needed on this aspect. For instance, the authors could re-use the very clear notation t, t+1, …, t+8 to define which time steps they use in terms of climate teleconnection indices with respect to the forecast month t.

We apologize for the misunderstanding. We used historical (t-8 to t-1) climate indices (ENSO, PDO, NAO and AMO). To clarify this point, we changed Lines 164-165 and updated Figure 1, as shown below:

*"Then, we estimate the lag-correlations between future monthly inflows over the next 7 months (t+1 to t+7) and historical climate indices (t-1 to t-8), snowfall (t to t-8), and inflow and soil moisture in current month (t)."*

[Figure]

*Figure 1.2. Graphical representation of the monthly prediction (MP) model scheme. At each calendar month t, we develop seven independent models to predict monthly inflows for the next seven months: MP1 (t+1), MP2 (t+2), ..., MP7 (t+7).*

Section 3.3.2: Even though the authors argue that MdAPE has a higher correlation and that it provides a value at each time step, KGE, and in particular its components, may have given insights into the forecast characteristics (correct timing, volumes, variations) that influence value. This information would be extremely valuable to guide further model and forecast developments for the hydropower sector, in the same way your investigation of dam characteristics informs dam managers of potential forecast value. I wonder whether this would be something to explore also to address the limitation you note in the Results section L.341 "For dams with poor IDF and high KGE, two features are noteworthy: first, KGE may not fully capture the relationship between forecast skill and value".

Thanks for pointing this out. We agree that KGE and its components may provide meaningful characteristics of forecasts, hence we looked into the correlation between these forecast characteristics and the performance metric *I* (Table 1.1). The correlation drops with longer lead-times, which is expected, suggesting that better prediction for immediate months tends to lead to higher forecast value. However, this trend is not observed for the bias ratio (beta), which suggests that accurate prediction in inflow volumes for all seven future months contributes to higher forecast value. Yet, the correlation values here are still lower compared to other indicators of forecast skill presented in Table S4 (e.g., correlation between *I* and MdAPE is -0.4). We think including these results in the Supplement may therefore be the best option (please refer to Text S5 and Table S5).

|       | MP1  | MP2  | MP3  | MP4  | MP5  | MP6  | MP7  |
|-------|------|------|------|------|------|------|------|
| KGE   | 0.21 | 0.15 | 0.14 | 0.11 | 0.09 | 0.07 | 0.06 |
| r     | 0.23 | 0.16 | 0.15 | 0.12 | 0.10 | 0.08 | 0.06 |
| beta  | 0.17 | 0.20 | 0.16 | 0.16 | 0.19 | 0.20 | 0.17 |
| gamma | 0.17 | 0.11 | 0.12 | 0.08 | 0.08 | 0.06 | 0.04 |

*Table 1.1. Correlation between performance metric I and forecast skill for the 269 dams that are classified as success cases. Forecast skill is represented by KGE and its three components, r (correlation), beta (bias ratio of mean inflow), and gamma (variability ratio). The columns correspond to the prediction model with 1 to 7 months lead-time.*

Forecasts with horizons up to 7 months are generally probabilistic to account for uncertainties at such long lead times. The authors should discuss the role of uncertainties in their study design, i.e. how realistic it is to consider the value of deterministic long-range forecasts depending on the current state of hydro-climate long-range forecasts, but also on the capacity for hydropower dam managers (whose actions are hypothesized in this study) to inform their decisions based on probabilistic information.

We agree this is a point worth discussing. We did so by expanding Section 5.2:

*"Finally, the investigation of alternative forecast approaches may be warranted. In particular, our deterministic long-range forecasts could be replaced by probabilistic ones, based for example on ensemble dynamical forecasts or statistical models including a stochastic representation of the residuals. By adopting probabilistic forecasts, one could represent additional operational aspects, such as the capacity of hydropower dam managers to inform their decisions based on probabilistic information. Such extension to our study could possibly uncover greater potential for improving dam operations (Zhao et al. 2011) and allow for a more nuanced quantification of forecast value."*

**Specific comments**

L.132-135 There is a range of models that fall between statistical prediction models and physically-based models. For instance, conceptual models do not fit in these two broad categories. I invite the authors to revise this statement.

We modified Lines 134-138 as follows:

*"Seasonal streamflow forecasting approaches include physically-based (mechanistic) models, such as GloFAS (a global-scale forecasting system; Emerton et al. (2018); Harrigan et al. (2020)), empirical or statistical (data-based) models that leverage the relationship between large-scale climate drivers and local hydro-meteorological processes (Block, 2011; Gelati et al., 2014; Giuliani et al., 2019), and conceptual (parametric) models that integrate hydrological processes at the catchment scale (Lindström et al., 2010; Devia et al., 2015)."*

L.137-140 The arguments for choosing a statistical model rather than a physically-based one seem too general. In fact, the statement "the prediction horizon of most physically-based

approaches (a few days to 3-4 months) falls short of our preferred lead times up to seven months" only holds when considering currently openly available global reforecasts, and not reforecasts from physically-based (or rainfall-runoff) models in general. There already exists, for instance, global reforecasts up to 7 months ahead and with hindcast periods for at least 30 years (https://hypeweb.smhi.se/explore-water/forecasts/seasonal-forecasts-global/). As the authors rightfully mention in the section on opportunities, "global-scale forecasting systems are gaining momentum", and therefore this part should be rewritten to highlight the impermanence of the statements.

*Thanks for the suggestion, which we reflected in Lines 143-150:*

*"Here, we select the second approach because of two reasons. First, the prediction horizon of most openly-available global re-forecasts (from a few days to 3-4 months) falls short of our preferred lead time (up to seven months), which is needed to test the potential of realistic forecasts for a broad spectrum of reservoirs—including those characterized by slow storage dynamics. Second, re-forecasts issued by global-scale forecasting systems are only available for a relatively-short hindcast period (typically two decades; Harrigan et al. (2020)), whereas the time series of globally-available hydro-climatological data are significantly longer. It should be noted that these two statements may change in the near future as the boundaries of global-scale forecasting systems keep getting extended (see Section 5.2). For example, there already exist global re-forecasts from physically-based models with a prediction horizon of seven months and hindcast periods of about 30 years (https://hypeweb.smhi.se/explore-water/forecasts/seasonal-forecasts-global/)."*

L.141 "Our long-range inflow prediction **model** uses…"

L.145 "For example, forecasts **issued**…"

*Thanks for spotting these two typos. We corrected them.*

L.174 Isn't it the goal of the dam inflow prediction model to feed the reservoir model? If so, isn't $Q_t$ not only retrieved from WaterGAP but also from the proposed statistical dam inflow model?

*At month $t$, the prediction model gives estimates of $Q_t$ to $Q_{t+6}$, which consequently determine the release sequence $R_t$ to $R_{t+6}$ (in Eq. 4). Then, only the decision $R_t$ is implemented. When simulating the reservoir dynamics, the observed inflow $Q_t$ retrieved from WaterGAP is used.*

Section 3.2.1 Is there any need for initialization of this reservoir model, and if so, how do you handle this aspect? e.g. which initial values do you use for instance for the reservoir storage?

*All reservoirs begin at full storage at the start of the simulation period (i.e., 1958). We clarified this point in Section 3.2.3.*

L.227 "… may **influence** ..."

L.234 "It is reasonable to hypothesize **that** the value…"

Thanks for spotting these typos. We corrected them.

L.251-253 "Note that failure implies that the control rules and perfect forecast-informed operations generate a similar amount of hydropower, meaning that information on storage and previous-month inflow are sufficient for near-optimal release decisions." Wouldn't that correspond to an $I_{PF}$ value of 0 rather than to the mean $I_{PF}$? If this statement is based on the mean $I_{PF}$ value, the reader does not have this information yet, and this sentence is confusing.

We agree that this statement is confusing, since 'failure' in this case actually means that the reservoir has $I_{PF}$ value < mean $I_{PF}$ which could be (and in most cases, is) greater than 0. We thus changed the term from "success/failure" to "case/non-case" and removed Lines 281-283 to avoid the confusion. The text has thus been changed as follows:

*"First, for each dam, we label it as case (also referred to as success) if it has the desired property of an $I_{PF}$ value larger than the mean value of $I_{PF}$ across all dams. Otherwise, the dam is labeled as non-case."*

L.320-321 "Considering the superior performance of the MP1 model, the forecast skill of MP1 only is retained to represent the overall forecast skill in the following analyses." Since the optimisation uses all forecast horizons, the speed with which skill decreases with the forecast horizon may play a role in the optimization and could have been considered as well.

Thank you for raising this point. It is indeed true that the speed with which forecast skill decreases may play a role. To investigate this, we first fit a linear regression between KGE and prediction lead time for each of the 269 dams classified as success cases. We then use the slope of the regression to represent the speed with which forecast skill decreases (i.e., a highly negative slope means forecast skill drops quickly with longer lead-times). We then plot the performance metric *I* against the slope. A shown in Figure 1.3, there is no clear trend of correlation between the speed with which forecast skill decreases and forecast value. Given this result, we preferred to report this analysis in the supplemental material (Text S6 and Figure S4).

[Figure]

*Figure 1.3. Scatter plot of performance metric I and slope of KGE against forecast lead time for the 269 dams classified as success cases. The blue line represents the local polynomial regression fitting performed on the data points (i.e., fit at point x is done using points in the neighborhood of point x).*

L.327-329 "Small negative values of $I_{PF}$ are likely a result of the discretization needed by dynamic programming to optimize the release sequence (eq. (4)), hence allowing control rules to outperform perfect forecast-informed operations." Could you please further explain what you mean to help understand the counter-intuitive negative $I_{PF}$ values?

Thank you for raising this point. First of all, this point made us realize that our explanation of the discretization process for stochastic and deterministic dynamic programming (for benchmark control rules and forecast-informed scheme) was not described with sufficient details. Such discretization is required when implementing dynamic programming algorithms, so we somewhat gave it for granted. However, we realize that many readers may not be familiar with this concept, which is now introduced in Lines 239-247 (when describing the experimental setup).

Having introduced the discretization process, we can now explain how it may affect the performance of forecast-informed operations. To begin the explanation, let's first of all consider that release decisions are discretized into 20 levels while storage is discretized into 500 levels. When the storage level falls between two discrete levels, the closer level is selected and the optimum release decision for that discrete level is implemented. This decision may sometimes be suboptimal, giving rise to negative $I_{PF}$ values. (To give an example in terms of specific values, let us consider a case in which the storage levels are discretized into 10, 20, 30, 40, etc. units of volume. Now, suppose that optimal release at storage = 30 is 1 unit volume and that optimal release at storage = 40 is 2 units of volume. Then, when the reservoir is at 34 units of volume, our policy would suggest releasing 1 unit of volume. However, the optimal decision at storage = 34 could be 2 units volume. This leads to suboptimality.) Note that the $I_{PF}$ values are small in

absolute terms and do not affect the interpretation of our results. We clarified this point in the revised version of the manuscript (please refer to Lines 363-365).

L355-357 "This is attributed to the weekly operations, suggesting that more frequent release decisions may reduce forecast value, since the benchmark operating rules have more opportunities to adjust release decisions." Isn't it the case for all the dams below the horizontal line? Why should these ones (the failing ones) behave differently?

Our original intention was to make a comparison between weekly operations and monthly operations. For all dams below the horizontal line, weekly operations reduce forecast value as shown in the figure below. We did not mean to make a comparison between the failing dams and the successful ones below the horizontal line. We realize that this statement can be misleading, hence we revised it to clarify our point:

*"This is because weekly operations decrease $I_{PF}$ for some of these dams to below the mean $I_{PF}$, turning them from cases (if operated on a monthly basis) into non-cases. This suggests that more frequent release decisions may reduce forecast value, since the benchmark operating rules have more opportunities to adjust release decisions."*

[Figure]

*Figure 1.4. Probability of success estimated using logistic regression when all dams adopt monthly operations (left) and when smaller dams (below the dashed line) adopt weekly operations (right, same as Figure 5 in the manuscript). Weekly operations tend to reduce forecast value as shown by the smaller blue circles and greater number of non-cases (triangles) in the right panel.*

L.446 "… a number of assumptions that must **be** properly contextualized."

Thanks for spotting this.

Figure 2 It would be more correct to change the caption to "Percentage of dams **whose inflow is** significantly correlated with…" since a dam in itself is not correlated to anything.

Revised as suggested.

Figures 3 and S2. The titles for the color scales in the second and third lines of this figure are confusing. If my understanding is correct, I would suggest changing titles in the first column to "Change in number of predicted months", and in the second column "Change in KGE", with "(b) MP4-MP1" and "(c) MP7-MP1" on the left-hand side.

Thanks for your suggestion. We modified both figures as suggested (please refer to the figures reported below).

Figure 5 "red triangles **represent** *failures*"

Figure 6 "meaning that the performance of realistic forecasts is worse **than** the one attained by control rules"

Thanks for spotting these typos. We corrected them.

**References**

*Knoben, W. J. M., Woods, R. A., and Freer, J. E.: A Quantitative Hydrological Climate Classification Evaluated With Independent Streamflow Data, Water Resources Research, 54, 5088–5109, https://doi.org/10.1029/2018WR022913, 2018.*

*Zhao, Tongtiegang, Ximing Cai, and Dawen Yang: Effect of streamflow forecast uncertainty on real-time reservoir operation, Advances in water resources 34.4 (2011): 495-504.*

[Figure]

*Figure 1.5. Number of months in which a predictive model is developed (left) and corresponding KGE (right). Taking a model with a lead-time of 1 month (MP1) as reference (a), we report the difference between MP1 and MP4 (b) and MP1 and MP7 (c).*

[Figure]

*Figure 1.6. MSESS (left) and GSS (right) values for 735 dams. Taking a model with a lead-time of 1 month (MP1) as reference (a), we report the difference between MP1 and MP4 (b) and MP1 and MP7 (c).*

**Reply to RC2: 'Comment on hess-2021-518', Anonymous Referee #2, 31 Jan 2022**

The paper contributes a global analysis about the value of long-term forecast for hydropower reservoirs. Specifically, the authors contrast the performance of three alternative operating schemes, basic control rules, perfect forecast-informed, and realistic forecast-informed. The latter use forecast information generated with a statistical prediction model based on four large-scale climate drivers along with local drivers (inflow and soil moisture). Results obtained for 735 hydropower reservoirs show that most dams could benefit from perfect forecasts, with these gains that strongly depend on dam characteristics; only a small number of dams however attains a performance improvement when realistic forecasts are used. The topic of the paper is absolutely timely and important, and fits nicely within HESS scope. The numerical analysis is robust and well designed, and the manuscript is clearly written. Overall, I think the paper could be a strong contribution to the ongoing debate about the relationship between forecast skill and value. Below I'm suggesting a few points to further improve the paper before accepting it for publication.

We thank the reviewer for the positive comments and further critical comments that we believe have enhanced the overall quality of the manuscript.

1) the description of the dam inflow prediction model in section 3.1 is not totally clear:

1a. I did not understand is the determination of the optimal set of lead-months at line 155. Does this mean that, for each station/HP reservoir, you constructed 7 forecasts (i.e. M1 to M7) and then selected the best lead-time as the one characterized by the minimum MSE? If this interpretation is correct, how can you then run a Model Predictive Control with a 7-month prediction horizon in case the best lead-time is shorter than 7? Moreover, since forecast accuracy generally decreases with lead-time, how likely will be then selecting a lead-time longer than one month?

We apologize for the misunderstanding. We constructed 7 forecast models (MP1 to MP7) for each dam, and the lead-month here refers to the lag-time of each predictor. For example, after we select four (statistically significant) predictors from lag-correlations, we choose an optimal combination of four lag-times of the predictors based on the minimum mean squared error. This applies to all 7 models independently and does not affect the prediction horizon. For clarifying this, Lines 171-173 have been changed to:

*"To select the optimal lag-times of the predictors, we apply a leave-one-out cross-validation (LOOCV) scheme. Specifically, all combinations of lag-times of the predictors are cross-validated; then, the optimal set of lag-times is determined based on the minimum mean squared error (MSE)."*

Regarding the forecast skill over lead-time, we found that the highest Kling-Gupta efficiency (KGE) appears in 68% of MP1, 5% of MP4, and 2% MP7 models. This is illustrated in Figure 3 and Figure 2S and explained in Lines 341-344.

1b. At line 142 the authors mention the generation of streamflow forecast for 1,200 stations, but the HP reservoirs are 735. Why are you generating a higher number of forecasts wrt the

reservoirs? Moreover, is it correct to say you built 1,200 independent forecast models, one for each station, right?

We apologize for the misunderstanding. The term "1,200 stations" refers to the prior study (Lee et al., 2018). Also, we built 7 independent MP models for each dam. To clarifying that, Lines 155-160 have been changed to:

*"Our long-range inflow prediction model uses Principal Component Regression (PCR) and includes four lagged large-scale climate drivers, snowfall, and prior inflow and soil moisture conditions to predict future inflows at 735 dams. This approach is readily implemented globally and has demonstrated fair (realistic) predictive skill at 1,200 streamflow stations (Lee et al., 2018). While Lee et al. (2018) predict seasonal (3-month) streamflow averages, here we develop independent monthly prediction (MP) models for the subsequent seven calendar months. For example, forecasts issued at the end of February include monthly inflows from March (MP1) to September (MP7)."*

1c. at line 138 you say that state-of-the-art physically-based forecasts fall short on lead-times up to 7 months, but actually these lead-times are covered by existing products such as ECMWF seasonal forecasts available on the Copernicus Data Store. I would thus recommend to better contextualize this point.

This is a point that was also raised by reviewer #1. We agree with this comment and have thus modified the first paragraph of Section 3.1 as follows:

*"Seasonal streamflow forecasting approaches include physically-based (mechanistic) models, such as GloFAS (a global-scale forecasting system; Emerton et al. (2018); Harrigan et al. (2020)), empirical or statistical (data-based) models that leverage the relationship between large-scale climate drivers and local hydro-meteorological processes (Block, 2011; Gelati et al., 2014; Giuliani et al., 2019), and conceptual (parametric) models that integrate hydrological processes at the catchment scale (Lindström et al., 2010; Devia et al., 2015). Here, we select the second approach for two reasons. First, the prediction horizon of most openly-available global reforecasts (from a few days to 3-4 months) falls short of our preferred lead times (up to seven months), needed to test the potential of realistic forecasts for a broad spectrum of reservoirs—including those characterized by slow storage dynamics. Second, re-forecasts issued by global-scale forecasting systems are only available for a relatively-short hindcast period (typically two decades; Harrigan et al. (2020)), whereas the time series of globally-available hydro-climatological data are significantly longer. It should be noted that these two statements may change in the near future as the boundaries of global-scale forecasting systems keep getting extended (see Section 5.2). For example, there already exist global re-forecasts from physically-based models with a prediction horizon of seven months and hindcast periods of about 30 years (https://hypeweb.smhi.se/explore-water/forecasts/seasonal-forecasts-global/)."*

2) The labeling of dams in success/failure (section 3.3.1) based on the comparison of IPF against the average IPF raises the following question: while the definition of IPF implies that forecast-informed operation is beneficial when IPF>0, I don't understand why a failure (i.e. IPF <

mean(IPF)) implies that basic control rules and perfect forecast-informed operations generate similar amounts of hydropower (lines 251-252). According to this condition, I guess a dam can be classified as failure even if IPF > 0, right?

We agree that this statement is confusing, since you rightly pointed out that a dam can be classified as 'failure' when its IPF value is positive. We thus changed the term from "success/failure" to "case/non-case" and removed Lines 281-283 to avoid the confusion.

3) While I fully trust the statistical forecast model used by the author, I think the paper could benefit from some benchmarking of the resulting forecast skill against existing, physically-based forecast products. This is likely not necessary for all the models, but it could be a useful complementary information for some representative cases, possibly selected across different climate regions.

We agree with the reviewer's point on a comparison of forecast skill with physically-based forecast products. However, there are two challenges that may hinder the comparison: 1) At the majority of dams, both our statistical model and physically-based prediction methods (or products) may predict different "simulated" streamflows rather than the actual "observed" streamflows. For instance, we used streamflow data predicted by the WaterGAP model. In other words, the result of such comparison may be affected by the different characteristics of streamflow simulations (e.g., forcing data). 2) The outcome of such analysis may vary significantly depending on the composition of the subset of skilled or unskilled regions. Other minor issues include obtaining data for the exact grids of dam locations, supporting finer spatial resolution for headwater dams, and forecasting the same time period with the same lead-time.

Because of these reasons, we believe a qualitative comparison may be the best choice. To this purpose, we retrieved the performance of the Global Flood Awareness System (GloFAS), one of the most advanced physically-based streamflow forecasting systems. Figure 2.1 shows the Kling–Gupta efficiency skill score (KGESS) for GloFAS-ERA5 river discharge reanalysis against 1,801 observation stations. While KGESS values are higher than the initial KGE values (Harrigan et al., 2020), the KGE scores generated in our study are comparable to or slightly higher than GloFAS scores (Figure 2.2). Even though the GloFAS's KGESS is calculated using observed streamflow, similar patterns of forecast skills can be found in our statistical forecasts, such as relatively lower forecast skills in central southern USA, southern South America, and southern East Africa, and relatively higher forecast skills in northwest North America, central South America, Europe, and South Asia.

Because of the reasons outlined above, we believe that adding such analysis to the revised version of the manuscript is not necessary, so we preferred to confine it to our response to the reviewers.

[Figure]

*Figure 2.1. Modified Kling–Gupta efficiency skill score (KGESS) for GloFAS-ERA5 river discharge reanalysis against 1,801 observation stations. Optimum value of KGESS is 1. Blue (red) dots show catchments with positive (negative) skill (Harrigan et al., 2020).*

[Figure]

*Figure 2.2. KGE scores of 1-month lead (MP1) inflow forecasts developed in our study. The original figure is Figure 3 in the manuscript.*

4) the results show how the overall value of forecast information for hydropower production is (unfortunately) relatively small. Did the author consider how much is the potential influence of the experimental settings, particularly in terms of (A) informing the operation with monthly inflow forecasts and (B) assuming the reservoirs are operated to maximize total (or average) hydropower production. About (A), the work by Bertoni et al. 2021 shows how some reservoirs could benefit more from predicting the inflow peak over a given horizon, rather than the average inflow, as this information is useful in hydropower operations to avoid spilling water. About (B), I

was wondering if in this context the maximization of the firm energy could benefit more than the maximization of total production as it is more related to extreme conditions.

Yes, these are two points that we considered when conceptualizing the study and setting up the experiments. However, we preferred to proceed with the current setup because of a few reasons. Starting with point (A), the nature and intent of a global study require us to create a realistic setup for all reservoirs of our study site. In this regard, it is true that some reservoirs could benefit more from predicting the inflow peak (instead of the total / average inflow volume), but investigating such aspect would result in a redesign of the study, which should bank on different forecast models and different analyses of how skill and dam design specifications result in forecast value. In other words, we preferred to keep a setup that is likely to reflect what the majority of dams would benefit from. As for point (B), the rationale is similar: we opted for a setup that is likely to reflect the operational objective characterizing most reservoirs. Such choice is corroborated by the validation reported in Turner et al. (2017), where we show that maximizing total production leads to an accurate simulation of annual hydropower production. That said, we agree with the reviewer that both point (A) and (B) are relevant to our study, so we expanded our reasoning in Section 5.2.

**MINOR:**

- in eq. 2c, the mass balance equation includes the evaporation losses. Where are these data coming from?

Evaporation is calculated by multiplying the surface area of a reservoir (at each time period) by the potential evaporation. Time series of potential evaporation from 1958 to 2000 are obtained from the Water and Global Change (WATCH) 20th century model output generated using the WaterGAP model (i.e., the same source as our time series for the inflow into each reservoir). We clarified this point in the revised version of the manuscript.

**References**

Lee, D., Ward, P., and Block, P.: Attribution of Large-Scale Climate Patterns to Seasonal Peak-Flow and Prospects for Prediction Globally, Water Resources Research, 54, 916–938, https://doi.org/10.1002/2017WR021205, 2018.

Turner, S. W., Ng, J. Y., & Galelli, S. (2017). Examining global electricity supply vulnerability to climate change using a high-fidelity hydropower dam model. Science of the Total Environment, 590, 663-675.

Harrigan, S., Zsoter, E., Alfieri, L., Prudhomme, C., Salamon, P., Wetterhall, F., Barnard, C., Cloke, H., and Pappenberger, F.: GloFAS-ERA5 operational global river discharge reanalysis 1979–present, Earth Syst. Sci. Data, 12, 2043–2060.